# Exploring the pharmacist's role in supporting newcomer international students and their families with the transition to the Canadian healthcare system including medication use: Protocol for a qualitative study

**Yasmin H. Aboelzahab**[1], **Andrew D. Pinto**[2,3,4,5], **Lisa McCarthy**[1,6], **Lisa Dolovich**[1]*

**1** Leslie Dan Faculty of Pharmacy, University of Toronto, Toronto, Ontario, Canada, **2** Upstream Lab, MAP Centre for Urban Health Solutions, Li Ka Shing Knowledge Institute, Unity Health Toronto, Toronto, Ontario, Canada, **3** Department of Family and Community Medicine, St. Michael's Hospital, Toronto, Ontario, Canada, **4** Department of Family and Community Medicine, Faculty of Medicine, University of Toronto, Toronto, Ontario, Canada, **5** Division of Clinical Public Health, Dalla Lana School of Public Health, University of Toronto, Toronto, Ontario, Canada, **6** Institute for Better Health, Trillium Health Partners, Mississauga, Ontario, Canada

* lisa.dolovich@utoronto.ca

**Data Availability Statement:** No datasets were generated or analyzed in the current protocol.

## Abstract

Canada welcomes a large number of newcomers every year including international students and their families. The over 800,000 international students who arrived in Canada in 2022, are twice as likely to face difficulties in accessing healthcare services compared to those who were born in Canada. Lack of access to healthcare services may jeopardize their full participation in Canadian society. Pharmacists are highly accessible healthcare providers due to their regular contact with the public, extended working hours, and wide distribution of pharmacies. Given the expanding role of pharmacists in Canada, pharmacists can be a great resource for international newcomers to address their healthcare needs when transitioning to the Canadian healthcare system and exploring available services. In this study, we will explore how international students and their families, who make up a significant segment of newcomers to Canada each year, transition to Canada's healthcare system and identify their main challenges related to developing coping skills for managing diseases and navigating the complexities of prescription and nonprescription medication use. To do so, we will conduct virtual interviews with newcomer international students who have lived experience with transitioning to the Canadian healthcare system. This study will help understand the international student experience and identify how pharmacists can deliver a tailored package of pharmaceutical services to this population to best address their healthcare needs when transitioning to the Canadian healthcare system with the goal of reducing healthcare gaps and inequities. By exploring the healthcare challenges faced by these students, our findings will help pinpoint precise areas where pharmacists can practice to their full scope including medication management, patient education, and transitional care. Consequently, the study will offer detailed recommendations on how pharmacists can better

Availability and access to data upon study completion will be in accordance with the guidelines of the University of Toronto Health Sciences Research Ethics Board guidelines.

**Funding:** This work is supported by Leslie Dan Faculty of Pharmacy, University of Toronto, and the Network for Improving Health Systems (NIHS) Trainee Award, University of Toronto. There was no additional external funding received for this study.

**Competing interests:** The authors have declared that no competing interests exist.

support this population and thereby help alleviate the overall strain on the primary healthcare system.

## Introduction

According to the Canada Health Act, providing reasonable access to healthcare services without barriers is a crucial determinant of good health [1]. Despite efforts exerted to equalize access to healthcare services, health status among different populations varies greatly, especially for Canada's newcomers [2]. The term "newcomer" refers to a person who has been in Canada for less than five years [3].

Canada's newcomers face challenges as they adjust to a new society [4]. In this respect, they face several barriers including insufficient understanding of the Canadian healthcare system, economic burdens, cultural differences, issues related to discrimination, language barriers, and mental health problems associated with trauma experienced in the country of departure [4, 5]. Other barriers relate to processes within the Canadian healthcare system such as accessibility and responsiveness of Canadian healthcare professionals in addressing the health needs of this population [6].

The health status of newcomers contributes to the overall health of the Canadian population due to the large number of newcomers in Canada [7]. Newcomers may suffer from communicable diseases, including respiratory and dermatologic infections, non-communicable chronic diseases, such as diabetes and hypertension, and other diseases preventable by vaccination [5]. If they are unable to access healthcare services when they are needed to manage and prevent illness, their health can be compromised which may also impact the general health of the Canadian population [4, 7].

A large portion of the newcomer population consists of international students [8]. The phenomenon of international student mobility is a global trend, driven by students' pursuit of educational and career opportunities across borders [8]. Each year, millions of students navigate complex and unfamiliar healthcare systems in their host countries, facing several barriers, and a general lack of knowledge about available healthcare resources [9–11]. Access to healthcare is crucial for international students, not only to prevent the exacerbation of health disparities but also to ensure their academic and social success. Without adequate healthcare services, international students are at a higher risk of facing untreated health issues, which can impede their ability to perform academically, integrate socially, and overall, succeed in their international educational endeavors [9].

Every year Canada welcomes a growing number of international students from over 200 countries around the world [12]. At the end of 2022, the international student population in Canada reached a total of 807,260, representing a significant segment of the country's newcomers [13]. While there are important benefits to furthering a diverse Canadian population that includes those new to Canada, rapid growth in newcomers has led to recent public policy changes to limit the number of newcomer international students as one mechanism to reduce national pressure on housing and everyday services including health care [14]. This measure reflects a deliberate shift towards a balanced approach, prioritizing both the well-being of international students and the capacity of local services to accommodate and integrate them effectively into Canadian society [14].

International students may face barriers while trying to access Canadian healthcare services including cost, community awareness, and cultural or religious sensitivity [12, 15]. Hindering their ability to access these services may jeopardize their full participation in Canadian society [16]. Transition to a new healthcare system is the main challenge international students face to stay healthy in Canada [15]. Compared to domestic students, international students are less likely to develop coping skills for managing diseases including the use of prescription and non-prescription medication [12] and the fear of getting sick due to a lack of information about treatment insurance coverage and medication dispensing policies that contribute to delays in treatment [15].

Moreover, newcomer international students are confused about how to obtain medications for different ailments and they do not fully understand information provided about the prescribed medications [15]. To overcome medication differences between their country of origin and Canada, international students may bring first aid kits and other medications that require a prescription in Canada as a safeguard. However, with lack of guidance, they are mostly unable to properly use these medications once in Canada [15].

Pharmacists are among the most accessible healthcare professionals in Canada due to their regular and expected contact with the public, extended working hours, and the widely distributed locations of community pharmacies, representing high geographic accessibility of pharmaceutical care services [17–19]. Moreover, they are drug experts who can optimize the pharmacotherapy outcomes of medications and minimize any potential adverse events [20]. In doing so, they improve medication management through addressing the unsuitable selection of therapies, improper dosing, and other medication-related problems [21].

The full scope of pharmacy practice includes patient-centered clinical services [19]. These services involve prescription renewals for chronic diseases, initiating treatment regimens, medication reviews, and injections to improve therapeutic outcomes for patients [17]. Pharmacists also provide virtual care services when in-person pharmacy visits are not a viable option [22]. Additionally, the community pharmacist's role in transitions of care between different healthcare settings has shown to decrease medication errors and ensure the continuity of care [23, 24]. Therefore, pharmacists' full scope of practice is well aligned with helping to meet the healthcare needs of newcomer international students, particularly when newcomers are not registered with a family physician and have inaccurate expectations or misinformation regarding the healthcare services available in Canada [25, 26]. Improving the newcomer international students' experience with care transitions and medication management can play a critical role in promoting better health outcomes for this population [7].

This study recognizes the increasing global mobility of international students and the crucial role healthcare systems play in ensuring their well-being and academic success [27]. As countries continue to attract students from diverse backgrounds, it is essential to understand and address the unique healthcare challenges faced by these students [9, 27]. By exploring the healthcare experiences of newcomer international students in Canada, this research aims to illuminate the broader patterns and obstacles that could be prevalent across other host countries around the world.

Our study aims to provide a comprehensive understanding of the unique lived experiences of international students. We will explore the healthcare needs of these students as they transition from healthcare systems in their countries of origin to Canada. Additionally, the study will identify the diverse challenges that these students face. It also seeks to uncover gaps in healthcare delivery that may hinder the continuity of care for this population and explore how pharmacists can better support newcomer international students and their families to facilitate their transition to a new healthcare environment.

## Materials and methods

### Study design

A qualitative study using in-depth interviews will be conducted to understand the experiences of newcomer international students and their families with the transition of healthcare between their countries of origin and Canada. In-depth interviews aim at exploring comprehensive perspectives, experiences, and motivations through detailed, open-ended interactions between the researcher and participant. Additionally, it emphasizes deep understanding by using semi-structured interview guide with probing questions that encourage participants to extensively express their thoughts and feelings [28, 29]. An exploratory descriptive qualitative approach will be used to describe the lived experiences of newcomer international students and their families and contextualize how they perceived the challenges during their transition to the Canadian healthcare system [30, 31].

The use of a qualitative exploratory descriptive methodology is a pragmatic approach to explore practical and effective solutions that could help identify appropriate strategies that best suit newcomer international students and their families [30]. The qualitative approach will provide a deeper understanding of the healthcare transition experiences faced by newcomer international students and their families, focusing on the meanings and views of the participants [32]. The exploratory aspect will examine the transition of healthcare for this population because limited data is available on this topic within the literature [31, 33]. The aim of the descriptive aspect in this study is to go beyond simple observation and data recording to gain insights and to provide a true picture of the transition of healthcare from the perspective of newcomer international students and their families [31, 33]. This involves deeply engaging with the data to interpret and understand the implications and experiences of participants, thereby enhancing our understanding of how they experience healthcare transitions at both personal and community levels [31, 33].

### Participants

**Recruitment.** Newcomer international students will be recruited through Discovery Pharmacy which is an academic not-for-profit accredited community pharmacy affiliated with the University of Toronto (U of T) and situated at the Leslie Dan Faculty of Pharmacy in Toronto, Canada. Discovery Pharmacy actively fosters research and education with a specific focus on advancing pharmacy practice, medication management, and healthcare delivery innovation.

For participants recruited from Discovery Pharmacy, the pharmacy team will be informed of the study aims and the inclusion criteria to determine who to invite to participate in the study. The pharmacy team will provide an initial introduction to the purpose and nature of the study to interested participants using a predetermined script. Contact information of interested participants will be shared with the research team who will contact those interested to present the study in greater detail. Participants will be informed that the pharmacy team is not involved in conducting the study and that the data provided will be de-identified to eliminate possible power dynamic issues between participants and the pharmacy team. The initial invitation to learn more about the study, carried out by the Discovery Pharmacy team, is necessary for recruitment to establish a trusted and familiar point of contact. Pharmacy staff members often have established relationships with customers or patients, and their involvement in the initial contact can enhance the credibility and comfort level for potential participants.

In addition, a study poster will be used to promote recruitment through other formal and informal networks on campus, including international student's advocacy groups at U of T

and social media platforms. Recruitment started on September 1st, 2023, and is planned to end by May 30th, 2024.

By diversifying our recruitment strategies, we aim to gather a more comprehensive range of experiences from international students, encompassing both those who are engaged with existing healthcare services and those who may be less integrated. This will enhance our study's capacity to capture a diverse array of challenges and experiences related to healthcare transitions among newcomer international students at U of T.

**Sampling strategy.** The target population for the study is newcomer international students who came to Canada, more than 1 month but less than 5 years ago. This timeframe is selected because newcomers are defined as those who have been in Canada for less than five years [3]. In addition, setting a minimum time of one month helps ensure that participants have had sufficient time to engage with the healthcare system, allowing them to reflect on their initial experiences and the challenges faced during their early adaptation period. The sample size will be approximately 20 international students which will be expected to produce a range of perspectives and reach thematic saturation. Throughout the study, there will be continuous reflection and interpretation of the data to identify the stage when data thematic saturation is reached. Thematic saturation refers to the pattern of repetition that emerges in the data, indicating a shared experience with no additional themes directly relevant to the research question or aims [34]. An estimated sample size is crucial for initial planning; however, it is important to note that the requirement for additional participants will be continually reassessed during the research process.

Participants will be recruited through purposeful sampling because it focuses on choosing the cases that provide a comprehensive understanding of the study's central problem and illuminating answers to interview questions [35]. We will make concerted efforts to include international students from diverse countries. This diversity will enable the collection of data from students who have experienced various healthcare systems, providing a richer base of information for identifying common challenges and effective strategies to overcome such challenges and developing effective strategies to address them.

For the purposeful sampling strategy, criterion sampling will be used to select participants who meet a pre-determined set of inclusion and exclusion criteria [35]. Inclusion/exclusion criteria will be verified through self-reported data. The criterion sampling strategy was chosen to select participants who can provide rich information to uncover crucial gaps in the current healthcare services that hinder the continuity of healthcare [35]. This may lead to useful insights into potential improvements needed to facilitate the transition of care for newcomer international students and their families. The pharmacy team will use inclusion and exclusion criteria to determine the participants eligible for the study. The inclusion criteria are: 1) participants who are students at U of T, currently residing in Ontario, Canada, 2) Participants who are 18 years or older, 3) time elapsed since coming to Canada more than one month and less than five years, and 4) participants who can speak English and can be understood by the interviewer. The exclusion criteria are: 1) participants who have a Canadian partner because they are less likely to have difficulty obtaining information about healthcare services in Canada from their partners, and 2) participants who self-assess when asked that they cannot communicate by phone or using internet-based platforms.

**Informed consent.** The consent form clearly states that participation in the study is completely voluntary. A copy of the study consent form will be provided to the participants by email. The consent form will be explained over a phone call with the research team, giving a chance for participant questions to be answered. In addition, to accommodate participants whose first language is not English, the research team will address any language-specific

questions or concerns to clear up any confusion and ensure a comprehensive understanding of the consent process before conducting the interview.

The participant can view and sign the written consent form using a Research Electronic Data Capture (REDCap) link which is an internet-based software platform developed to facilitate the collection of data for research surveys in a secure manner. This can be done through scanning a QR code provided in the study poster or through clicking a direct link sent via email. Alternatively, a copy of the study consent form will be provided to participants by email or mail according to their preference. The participant can return the signed hard copy by email or mail in a prepaid envelope included in the participant consent form package.

The terms in the consent will be reviewed verbally before starting the interview. This review process will include a thorough explanation of the study's purpose, procedures, and participant rights to ensure complete understanding. Participants will have the opportunity to ask questions and seek clarifications, allowing them to make an informed decision about their involvement. This step is crucial for maintaining ethical standards and ensuring that all participants are fully aware of what participation entails.

## Data collection procedures

The interviews will be conducted virtually using an online platform (Microsoft Teams application). The use of Teams will eliminate any exposure to COVID-19 or other infections through in-person contact and provide an effective mechanism to record and transcribe the interview. Using an online platform will also increase the accessibility of the study to allow for more flexibility in scheduling and location for the interviews. Some participants may experience some challenges accessing private and confidential spaces for connecting to virtual platforms. Therefore, using the telephone to dial in and join the Microsoft Teams meeting will be offered to maximize the study inclusion and equity as much as possible. In doing so, participants who have limited access to private online platforms will be fairly accommodated in the study.

At the beginning of the interview, the interviewer will collect the interviewee's consent for the discussion to be recorded and transcribed via Microsoft Teams application. The generated transcripts will be reviewed, de-identified, and stored electronically on a secure network affiliated with the university. Any personal identifiers (names of individuals, locations, organizations, etc.) will be removed from the transcripts. The interview recordings will be securely stored and kept until transcription and analysis are completed. Once these processes are finalized, all the recordings will be permanently deleted to ensure participants' privacy and confidentiality are maintained in accordance with ethical research guidelines.

Interviews will be video recorded; however, participants will be advised that they can turn off their cameras if they want their interviews to be only audio recorded. Participants have the option to refrain from answering any questions they wish during the interview if they feel uncomfortable.

**Interview method.**   Understanding the newcomer international student's perspective is critical for uncovering various aspects of the transition of care between the healthcare system in the country of origin and the Canadian healthcare system. Hence, in-depth interviews will be conducted to generate meaningful experiences based on the interviewees' perceptions of the transition of care for newcomer international students and their families [36]. As sensitive matters may be raised, in-depth interviews will be conducted rather than focus groups to provide a private setting for participants to freely share their thoughts without violating their privacy or fear of judgment [37].

Participants will select their preferred location for conducting the virtual interview. The duration of each interview will be up to 60 minutes. The interview will start with an

introduction between the interviewer and the participant. This will be followed by explaining the study and its main goals, which will help focus the participants' answers on the study topic and avoid potential distractions.

An interview guide will be followed with the main questions that cover the topic of the study and potential follow-up questions based on the participant's responses to generate more information and encourage the participants to elaborate on their experiences (S1 Appendix). However, while all topics set out in the interview guide will be addressed, not all the questions included in the interview guide will be asked in each interview, depending on the flow of the interview and each participant's responses. This will allow the participants to take their time answering each question without rushing to the following questions, which can result in more in-depth data.

**Interview questions.** Participants' demographic data will be collected in the interview, including age, gender, date of arrival to Canada, and whether participants live alone or not. The semi-structured interview guide consists of 5 open-ended main and probing questions. Main questions seek to understand the experiences of newcomer international students and their families with the transition of health care between their country of origin and Canada. The questions also explore the main challenges during this transition state and the role of pharmacists in bridging effective and continuous transition of care and improving medication management for this population. In addition, participants will be asked questions about the current healthcare services they have been using to manage their health conditions. Specifically, they will be prompted to discuss their perceptions of how these services have met their healthcare needs, including any improvements in their conditions and satisfaction with the service received. A final question will be directed to ask about their suggestions on how to improve their experience with transitioning to the Canadian healthcare system and improve their health status in Canada. To ensure the clarity and effectiveness of these questions, the interview guide will be pilot tested with a small group of participants (2–3). This pilot test will help refine the questions to better capture comprehensive and relevant data.

## Data analysis and reporting

There is little known in the literature about the transition of healthcare for newcomer international students and their families [38, 39]. Therefore, conventional content analysis will be used to analyze the qualitative study data [40]. The data analysis will be conducted in parallel with the interviews to continually revise and update the interview guide per the responses obtained during the interviews. Updating the interview guide will involve only minor changes, such as including additional probes or modifying the language of some questions if they are found to be misunderstood.

The coding will be conducted by two researchers, each will be working separately on a subset of the interview transcripts. This dual-coding approach is essential for assessing the consistency between their analyses, ensuring that the analysis is unbiased and mutually exclusive, and enriching the data quality.

Transcripts will be analyzed using NVivo® 12 Plus software [41]. The first step of data analysis is to achieve immersion and assimilation of the whole data through iterative reading of the interviews' transcripts. The following step in the analysis will involve a thorough reading to start generating the codes that capture the main themes and highlight the exact text reflecting these themes. Then, notes about their ideas and perspectives on the initial step of the analysis will be taken. Next, an initial coding scheme will be developed by joining the code labels that capture more than one main theme [40].

Based on the linkage between different codes, categories will be formed. These categories are needed to sort the codes into conceptual clusters. After coding all the interview transcripts, the final codes will be examined and organized into a tree diagram which is a hierarchical structure. Examples of each code and category will be included in the study results [40].

The study findings will be centered and shared using several quotes from different participants to increase the transparency of the study and to support the researchers' interpretation of the collected data. The quotes will be edited and defined using appropriate identifiers such as the code name and the main characteristics [42].

The Consolidated Criteria for Reporting Qualitative Research (COREQ) checklist (S2 Appendix) will be used as an integrated framework for planning and reporting the study as it is suitable for studies using interviews [43]. It comprises 32 items to assess 3 main domains of the study including the research team, the study methodology, and the data analysis and interpretation. Using the COREQ checklist will ensure high-quality reporting of finding, increase the trustworthiness, and promote explicit and comprehensive reporting of the study [43, 44]. Additionally, it will cover the important role of researchers in the qualitative study by adding the context of their interpretation of the study findings and the assumptions they may have made. This will lead to increasing the study's transparency [43, 44]. A summary of 1–2 pages of the study findings will be shared with participants upon their request.

## Data management

Interview transcripts will be cleaned from misspellings, reviewed, and de-identified within a week after the interview time. To de-identify data, each participant will be assigned a unique identifier. The key linking participant identities to their unique identifier will be password protected and stored in a separate folder on the U of T server. The link between the individual and the recording will be maintained until the data analysis process is complete. Therefore, the data will be anonymized once the data analysis process is complete. Transcripts and interview recordings will be stored electronically on a secure network using SharePoint behind U of T server. The interview recordings will be kept until transcription and analysis are completed.

## Ethical considerations

This study was approved by the University of Toronto Health Sciences Research Ethics Board (REB # 44770). Participants are required to consent to the study and complete the study interview. They must provide informed consent and have any inquiries addressed. Participants have the option to refuse the recording of the interview and withdraw their consent from the study at any point during the interview or until their transcript has been de-identified. If a participant chooses to withdraw from the study, any collected data will be discarded upon their request.

Due to the study topic that covers potential negative experiences associated with challenges related to the transition of health care between their country of origin and Canada, the participants may become emotional while talking about these feelings. This could cause them to feel discomfort while sharing these negative personal experiences. The participants' mental well-being will be prioritized over the study procedures. Therefore, the participants will be informed at the beginning of the interview of their right to stop the recording, skip any question or withdraw from the interview. Additionally, while we are unable to provide direct therapeutic services, we will offer a list of accessible mental health resources and support services available to them as international students at U of T, ensuring they have options to seek help if needed. This approach ensures the ethical handling of emotional responses while respecting the participants' emotional well-being.

## Discussion

This study will help better understand the lived experiences of newcomer international students and their families. It will also aid in uncovering the gaps in the current healthcare services that hinder the continuity of healthcare. In this study, newcomer international students will have the chance to be actively involved in addressing their needs by sharing their experiences, providing their feedback on the current healthcare services infrastructure, and their suggestions on improving these services. In doing so, international students and their families will have a voice to communicate their needs and the message policymakers need to hear.

International students, while often perceived as privileged, face unique challenges and stressors related to academic pressures, cultural adaptation, and in some cases, the complexities of navigating a foreign healthcare system [39, 45]. These factors can contribute to specific health vulnerabilities that differ from those of other newcomer groups and the general population [12, 15, 39]. The insights gained from studying this particular subgroup can provide valuable information on the integration processes and the effectiveness of current healthcare services tailored to address the healthcare needs of newcomers including non-permanent residents in Canada [38]. Understanding these aspects can help identify gaps in services and inform policies that enhance healthcare accessibility and equity not only for international students but also for other newcomer groups. International students often form a bridge between their communities and the host country, influencing both cultural exchange and economic ties [15]. Studying their healthcare experiences can thus offer broader insights into how health systems and policies affect an internationally mobile population, which can have implications for global health policies and international collaboration [46]. Therefore, while the study concentrates on international students, the findings can illuminate aspects of the healthcare system's functionality and responsiveness, offering perspectives that are applicable to the broader discussion on healthcare accessibility and equity for all newcomers [46]. This approach allows for a focused investigation that can still contribute to understanding and addressing the wider challenges faced by newcomers.

Additionally, the findings from this research could reveal broader opportunities for improving healthcare access and quality, not just through pharmacists but through a more integrated approach involving other healthcare professionals [25]. Understanding how international students navigate healthcare challenges could lead to improvements in service delivery across a variety of healthcare settings, including primary care facilities and mental health services [45]. This could include better coordination between healthcare disciplines, tailored support systems for international students, and facilitated access to essential services, ensuring that all healthcare providers are better equipped to meet the diverse needs of this group effectively.

The study findings can also guide development and implementation of how pharmacists can deliver a tailored package of pharmaceutical services to this population which will facilitate their transition to the Canadian healthcare system, improve medication management, and reduce healthcare gaps and inequities [17, 25]. This will help provide new avenues for pharmacists' professional development and alleviating the strain on the primary healthcare system [19]. Importantly, the findings acquired from this study can be potentially expanded to other newcomer populations who may have similar needs associated with transitioning to the Canadian healthcare system.

Moreover, the insights from this study on the transition of newcomer international students to the Canadian healthcare system could be highly relevant to other countries with significant international student populations. Given the shared challenges these countries face, the findings and recommendations from this study can be adapted to enhance healthcare access for

international students worldwide. The barriers to healthcare access encountered by international students are not unique to Canada but are potentially common across other host countries [9, 27]. By identifying these challenges within the Canadian context, this study will provide valuable insights that could be tailored by other nations to address similar issues. Such adaptations are crucial for countries with large international student populations facing comparable healthcare access challenges. This approach can broaden our understanding of global healthcare needs for international students, highlighting both commonalities and variations.

The study has some anticipated limitations. The target population of this study is newcomer international students at the U of T and their families. As a result, the collected data may only reflect the experiences of a population that has moved to Canada in stable conditions and who has a high level of health literacy. Consequently, the data may not reflect the perspectives of other populations, such as refugees and illegal immigrants, who may have experienced more difficulties during their health transition to Canada or may have low health literacy levels. Some participants may have had very limited experience with the Canadian healthcare system so far and so may have difficulty identifying how their health needs can be met by an unfamiliar system.

U of T continuously welcomes international students from a diverse array of countries. According to the latest information provided by the university, more than 25% of the student body is international, coming from 170 different countries [47, 48]. The study sample includes approximately 20 participants. Consequently, our study emphasizes detailed understanding from these participants rather than broad applicability to all international students at U of T. The study findings may therefore reflect the experiences of international students from only the represented foreign countries. Our goal is to deeply understand the specific experiences and challenges faced by the students who are part of the study, recognizing that these insights are representative of only those individuals' experiences. Thus, while the study provides valuable insights into the experiences of its participants, these findings are understood to be specific to the context and individuals studied, offering detailed understanding rather than broad applicability.

Given the extensive prevalence of mental health challenges in young adults in university and the added challenges newcomer students face when coming to study in a new country [12], there is high interest in how to support newcomer students at U of T and indeed universities worldwide [39]. For example, initiatives such as those led by the Centre for International Experience (CIE) at U of T play a pivotal role by offering diverse programs and services designed to support international students [49] and International Students Health and Wellness service that focuses on the well-being of international students [50]. Other universities in Canada have similar structures and services to support international students.

In addition to university-led initiatives, the government of Canada offers Newcomer Services to facilitate a smooth transition for newcomers into everyday life in Canada [51]. These services provide valuable resources for job searching, housing, healthcare access, language skills improvement, and other essential services [51]. However, these initiatives may not provide sufficient coverage for the healthcare needs of newcomer international students. Therefore, leveraging community pharmacy teams with their accessibility, wide geographic distribution, and full scope of practice is a helpful additional strategy to contribute to health care delivery, and to support newcomer students to Canada [17–19]. It is expected that this work will be shared with academic, policy maker and practitioner audiences through a variety of outputs including academic presentations, posters and peer-review publications, policy briefs and education programs for practicing pharmacists.

The findings from this study are expected to be of interest to various stakeholders involved in healthcare policy, administration, and service delivery, particularly those focused on the

well-being of newcomer international students in Canada. Understanding the barriers, such as cultural differences, economic burdens, and language challenges, can also inform the development of policies and interventions aimed at improving healthcare equity and inclusivity for this population. Additionally, the study's focus on the role of pharmacists as accessible healthcare providers aligns with the expanding scope of pharmacy practice in Canada [19]. The findings are expected to be valuable for pharmacists seeking to enhance their role in addressing the healthcare needs of international students and their families during their transition to the Canadian healthcare system. Moreover, the study's outcomes will also have applications for educational institutions, community organizations, and advocacy groups involved in supporting the integration of international students into Canadian society, contributing to a more comprehensive and effective healthcare system for Canada's newcomers.

Overall, our study is a building block that provides foundational insights to support future research and health system reform that will bridge healthcare gaps, encourage professional development among pharmacists, and alleviate pressure on the primary healthcare system by promoting tailored and accessible pharmaceutical services for newcomer international students.

## Supporting information

**S1 Appendix. The interview guide.**
(PDF)

**S2 Appendix. Consolidated criteria for reporting qualitative studies (COREQ): 32-item checklist.**
(PDF)

## Author Contributions

**Conceptualization:** Yasmin H. Aboelzahab, Lisa Dolovich.

**Funding acquisition:** Yasmin H. Aboelzahab, Lisa Dolovich.

**Investigation:** Yasmin H. Aboelzahab, Lisa Dolovich.

**Methodology:** Yasmin H. Aboelzahab, Lisa McCarthy, Lisa Dolovich.

**Supervision:** Andrew D. Pinto, Lisa McCarthy, Lisa Dolovich.

**Validation:** Yasmin H. Aboelzahab.

**Writing – original draft:** Yasmin H. Aboelzahab.

**Writing – review & editing:** Yasmin H. Aboelzahab, Andrew D. Pinto, Lisa McCarthy, Lisa Dolovich.

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
