## [Decision Letter · Decision Letter 0]

3 Apr 2024

PONE-D-24-06002Exploring the pharmacist’s role in supporting newcomer international students and their families with the transition to the Canadian healthcare system including medication use: Protocol for a qualitative studyPLOS ONE

Dear Dr. Aboelzahab,

Thank you for submitting your manuscript to PLOS ONE. After careful consideration, we feel that it has merit but does not fully meet PLOS ONE’s publication criteria as it currently stands. Therefore, we invite you to submit a revised version of the manuscript that addresses the points raised during the review process.

I enjoyed reading your paper. The paper tries to explores the challenges faced by international students in Canada. However, the protocol can be equally important to other countries who are receiving international students. Therefore, I request you to add more relevant information in background, methodology and discussion section so that the protocol can be of wider use for international scientific communities. I also eco with reviewer's comments, please edit your paper as per the reviewer's comments. 

We look forward to receiving your revised manuscript.

Kind regards,

Kanchan Thapa, MPH, MPhil

Academic Editor

PLOS ONE

Journal Requirements:

“This work is supported by Leslie Dan Faculty of Pharmacy, University of Toronto, and the Network for Improving Health Systems (NIHS) Trainee Award, University of Toronto”

“This work is supported by Leslie Dan Faculty of Pharmacy, University of Toronto, and the Network for Improving Health Systems (NIHS) Trainee Award, University of Toronto.”

“This work is supported by Leslie Dan Faculty of Pharmacy, University of Toronto, and the Network for Improving Health Systems (NIHS) Trainee Award, University of Toronto”

4. Please be informed that funding information should not appear in the Acknowledgments section or other areas of your manuscript. We will only publish funding information present in the Funding Statement section of the online submission form. Please remove any funding-related text from the manuscript.

Additional Editor Comments (if provided):

Dear Authors,

I enjoyed reading your paper. The paper tries to explores the challenges faced by international students in Canada. However, the protocol can be equally important to other countries who are receiving international students. Therefore, I request you to add more relevant information in background, methodology and discussion section.

At this stage, I have following comments:-

Background should contain some of the information about global trends of students migration, how they face health care challenges, they provide an overview of Canada. Why it is important to have access to health care services, due to lack of which how is the situation in global, in regional and in Canada?

Aims and objectives can be stated as a paragraph in the last portion of introduction section.

Participants:-

Are the participants recruited from two institutions are generalizable about the situation of Canada?

Authors contribution section- please state as per the plos one format. Ensure all the authors mention the required criteria for authorship.

Reviewers' comments:

Reviewer's Responses to Questions

**Comments to the Author**

1. Does the manuscript provide a valid rationale for the proposed study, with clearly identified and justified research questions?

Reviewer #1: Partly

Reviewer #2: Yes

Reviewer #3: Yes

2. Is the protocol technically sound and planned in a manner that will lead to a meaningful outcome and allow testing the stated hypotheses?

Reviewer #1: Yes

Reviewer #2: Partly

Reviewer #3: Yes

3. Is the methodology feasible and described in sufficient detail to allow the work to be replicable?

Reviewer #1: Yes

Reviewer #2: Yes

Reviewer #3: Yes

4. Have the authors described where all data underlying the findings will be made available when the study is complete?

Reviewer #1: Yes

Reviewer #2: Yes

Reviewer #3: Yes

5. Is the manuscript presented in an intelligible fashion and written in standard English?

Reviewer #1: Yes

Reviewer #2: Yes

Reviewer #3: Yes

6. Review Comments to the Author

You may also provide optional suggestions and comments to authors that they might find helpful in planning their study.

Reviewer #1: This protocol is for a qualitative study that will explore the role that pharmacists could play in helping international students and their families during their transition from the healthcare services of their country of origin to the Canadian one. The following points will help strengthen the manuscript.

1. Lines 56-111: How do new rules for international students in Canada affect this protocol/study?

2. Lines 113-117: How is aim #1 different from aim #s 2 & 3? Please discuss.

3. Lines 123-125: Please provide a definition of “in-depth interviews”. Aren’t the interviews in a qualitative study already aim to explore ‘in-depth’ experiences of responders? What does the addition of “in-depth” mean here?

4. Lines 125-136: Thanks for providing references and explanations for “exploratory descriptive qualitative approach”. The qualitative approach is usually used to ‘explore’ and ‘describe’ phenomenon where there is little information available. Perhaps, authors should define what they mean by a “qualitative study” first for readership to understand the following sections about “exploratory” and “descriptive”. Further, as authors indicate, the “descriptive … will go beyond simple observation and data recording…”. However, authors do not provide what are the qualifications of the state that will be achieved “beyond…”. Please discuss.

5. Lines 198-211: Could authors please discuss the types of guarantees (in terms of the privacy and confidentiality of interview data) that are provided by Microsoft Teams.

6. Lines 208-210: Please discuss if authors have concerns about how comparable the interview responses will be when camera is on as opposed to when camera is off.

7. Lines 213-231: Who will conduct the interviews? Later, it is reported that there will be at least one interviewer (line 365). Please discuss if there will be other interviewers.

8. Lines 213-231: Will there be debriefing after interviews? Authors do discuss revising the guide later but that seems to be only tied to the data analysis. Please specify.

9. Lines 233-243: Will the interview guide be pilot tested?

10. Lines 234-237: “Semi-structured” interviews? Please tie back to “in-depth” interviews.

11. Lines 245-272: How will authors ensure that their results are credible?

12. Lines 245-255: Who will do the coding and other data analyses and how? There seems to be one interviewer (line365). Does that mean there will be one coder? Please describe the process clearly.

13. Lines 264-272: “COREQ”? Thanks for providing a reference but please ensure that the acronym is written out the first time it is mentioned.

14. Lines 288-294: Will there be any services available for those international students who become emotional during interviews?

15. Line 298: Again, “explorative descriptive qualitative study”?

16. Lines 323-327: Could authors explore which countries (of that total 200) are represented by U of T international students rather than reporting this as a limitation?

17. Lines 323-324: Please refrain from discussions that resemble (or make the reader think of) generalizability and give the impression that this protocol is for a quantitative study.

18. Lines 324-327: Again, which countries are those? Further, even if U of T had international students from all of those 200 countries, which characteristics of a qualitative study provide the researcher with the tools that will make the generalizability of their findings possible? Please discuss.

Reviewer #2: Dear Author,

I enjoyed the manuscript and found it informative.

After going through the manuscript and reviewing it, I found that there are some changes that need to be addressed to enhance the overall quality of the manuscript. Please find my comments on the attached file.

Reviewer #3: This is an excellent and novel protocol that holds promise in shedding light on an important and under-researched topic in Canada. The methods are clearly described, and the rationale for the study is also clear. I have only three main comments for consideration.

First, though the motivation for your study is about the issues facing newcomers in terms of access to care and inequities in health outcomes, the focus on international students may limit the ability to generate insights into the challenges facing newcomers more broadly, especially those in more structurally disadvantaged positions. Indeed, international students may be among the more privileged and may have advantages often described as the healthy immigrant effect, whereby levels of health on average for this population are better than for average Canadians, or newcomers more broadly. Could the authors elaborate on the importance of this particular subgroup of the newcomers population and how insights gained from this study could still be informative in terms of the broader issues presented in the introduction?

Second, it would help to clarify the recruitment a bit more, as it strikes me that the use of a pharmacy to recruit international students may select only those who already have some prior knowledge of existing care options available to them and who are able to access it vs those who are less aware of on campus services and supports and therefore may be facing greater barriers to care that are worth uncovering in the study. It would be great if the authors could comment on this potential limitation, or correct my misunderstanding.

Third, I understand that the motivation for this study is to examine the potential role of pharmacists in assisting navigation and transition to Canada's health systems for newcomers, but is there also room to uncover opportunities for other health providers, and perhaps gain insights into the opportunities to improve access or quality of care for this population by other care providers, such as primary care physicians, nurses, walk-in clinics, mental health services, etc? The broader implications of your study could be considered in the Discussion section, since the open-ended nature of the questions asked in the interview guide may yield new insights into these other aspects of health system.

7. PLOS authors have the option to publish the peer review history of their article (what does this mean?). If published, this will include your full peer review and any attached files.

Reviewer #1: No

Reviewer #2: **Yes: **Rita Adhikari

Reviewer #3: **Yes: **Sara Allin

---

## [Author Response · Author response to Decision Letter 0]

10 May 2024

Editor Comments:

Thank you for stating in your Funding Statement:

“This work is supported by Leslie Dan Faculty of Pharmacy, University of Toronto, and the Network for Improving Health Systems (NIHS) Trainee Award, University of Toronto”

Updated Funding Statement: “This work is supported by Leslie Dan Faculty of Pharmacy, University of Toronto, and the Network for Improving Health Systems (NIHS) Trainee Award, University of Toronto. There was no additional external funding received for this study.”

Dear Authors,

I enjoyed reading your paper. The paper tries to explores the challenges faced by international students in Canada. However, the protocol can be equally important to other countries who are receiving international students. Therefore, I request you to add more relevant information in background, methodology and discussion section.

Response to Editor

We would like to thank the editor for this comment. In response, we added more relevant information to several sections of our manuscript as follows:

- Introduction section: “This study recognizes the increasing global mobility of international students and the crucial role healthcare systems play in ensuring their well-being and academic success[27]. As countries continue to attract students from diverse backgrounds, it is essential to understand and address the unique healthcare challenges faced by these students[9,27]. By exploring the healthcare experiences of newcomer international students in Canada, this research aims to illuminate the broader patterns and obstacles that could be prevalent across other host countries around the world.”

- Methodology section: “We will make concerted efforts to include international students from diverse countries. This diversity will enable the collection of data from students who have experienced various healthcare systems, providing a richer base of information for identifying common challenges and developing effective strategies to address them.”

- Discussion section: “Moreover, the insights from this study on the transition of newcomer international students to the Canadian healthcare system could be highly relevant to other countries with significant international student populations. Given the shared challenges these countries face, the findings and recommendations from this study can be adapted to enhance healthcare access for international students worldwide. The barriers to healthcare access encountered by international students are not unique to Canada but are potentially common across other host countries[9,27]. By identifying these challenges within the Canadian context, this study will provide valuable insights that could be tailored by other nations to address similar issues. Such adaptations are crucial for countries with large international student populations facing comparable healthcare access challenges. This approach can broaden our understanding of global healthcare needs for international students, highlighting both commonalities and variations.”

At this stage, I have following comments:

Background should contain some of the information about global trends of students migration, how they face health care challenges, they provide an overview of Canada. Why it is important to have access to health care services, due to lack of which how is the situation in global, in regional and in Canada?

We added a section that briefly addresses global trends of students’ migration.

Introduction section: “A large portion of the newcomer population consists of international students[8]. The phenomenon of international student mobility is a global trend, driven by students' pursuit of educational and career opportunities across borders[8]. Each year, millions of students navigate complex and unfamiliar healthcare systems in their host countries, facing several barriers, and a general lack of knowledge about available healthcare resources[9–11]. Access to healthcare is crucial for international students, not only to prevent the exacerbation of health disparities but also to ensure their academic and social success. Without adequate healthcare services, international students are at a higher risk of facing untreated health issues, which can impede their ability to perform academically, integrate socially, and overall, succeed in their international educational endeavors [9].”

Aims and objectives can be stated as a paragraph in the last portion of introduction section.

We added the study aims and objectives as a paragraph in the last portion of the introduction section: “Our study aims to provide a comprehensive understanding of the unique lived experiences of international students. We will explore the healthcare needs of these students as they transition from healthcare systems in their countries of origin to Canada. Additionally, the study will identify the diverse challenges that these students face. It also seeks to uncover gaps in healthcare delivery that may hinder the continuity of care for this population and explore how pharmacists can better support newcomer international students and their families to facilitate their transition to a new healthcare environment.”

Participants:-

Are the participants recruited from two institutions are generalizable about the situation of Canada?

We would like to thank the editor for raising this important question. Our proposed study aligns with the aims of qualitative research, which prioritize depth and detailed representation rather than generalizability. Therefore, our study focuses on the depth, richness, and complexity of understanding from a limited number of cases or participants. 

In response, we added to Discussion section: “Our goal is to deeply understand the specific experiences and challenges faced by the students who are part of the study, recognizing that these insights are representative of only those individuals' experiences. Thus, while the study provides valuable insights into the experiences of its participants, these findings are understood to be specific to the context and individuals studied, offering detailed understanding rather than broad applicability.”

Reviewer #1: 

1. Lines 56-111: How do new rules for international students in Canada affect this protocol/study?

We would like to thank the reviewer for highlighting this important point. To address this question, we added in the introduction section: “While there are important benefits to furthering a diverse Canadian population that includes those new to Canada, rapid growth in newcomers has led to recent public policy changes to limit the number of newcomer international students as one mechanism to reduce national pressure on housing and everyday services including health care[14]. This measure reflects a deliberate shift towards a balanced approach, prioritizing both the well-being of international students and the capacity of local services to accommodate and integrate them effectively into Canadian society[14].”

2. Lines 113-117: How is aim #1 different from aims #2 & 3? Please discuss.

Aim #1 of the study focuses on the personal and subjective experiences of individuals as they navigate a new healthcare environment. It covers a wide range of experiences, including the initial understanding and adaptation to Canadian healthcare practices, cultural adjustments, and the emotional and psychological impacts of transitioning between healthcare systems.

Aim #2, on the other hand, narrows the focus specifically to the challenges associated with medication use and management. This involves exploring specific issues such as understanding prescriptions, accessing medications, dealing with differences in medication availability or regulations, and managing ongoing treatments that were started in their home countries. This aim looks at practical and logistical aspects of medication management that international students and their families face in the Canadian context.

Aim #3 extends the investigation into the structural and systemic aspects by identifying gaps in the healthcare delivery system that may affect the continuity of care for these newcomers. This could include gaps in service provision, barriers to access (like language barriers or lack of information), and inadequacies in policy that may disrupt the seamless continuation of healthcare services as students move from one healthcare system to another.

These details were not added into the paper because we also received a comment to ensure aims are concise.

3. Lines 123-125: Please provide a definition of “in-depth interviews”. Aren’t the interviews in a qualitative study already aim to explore ‘in-depth’ experiences of responders? What does the addition of “in-depth” mean here?

The addition of "in-depth" distinguishes these interviews from other qualitative methods like focus groups or structured interviews, where the scope for individual elaboration might be more limited due to the structured nature of questions or the group setting. In-depth interviews are particularly valuable in exploring complex issues, understanding personal stories, and gaining insight into contexts and behaviors that are not immediately observable. This method allows for a comprehensive exploration of the subject matter, which is essential in capturing the essence of participants' lived experiences and the meanings they attribute to them.

To better define the “in-depth interviews”, we added in Study Design section: “In-depth interviews aim at exploring comprehensive perspectives, experiences, and motivations through detailed, open-ended interactions between the researcher and participant. Additionally, it emphasizes deep understanding by using semi-structured interview guide with probing questions that encourage participants to extensively express their thoughts and feelings [28,29].”

4. Lines 125-136: Thanks for providing references and explanations for “exploratory descriptive qualitative approach”. The qualitative approach is usually used to ‘explore’ and ‘describe’ phenomenon where there is little information available. Perhaps, authors should define what they mean by a “qualitative study” first for readership to understand the following sections about “exploratory” and “descriptive”. 

Further, as authors indicate, the “descriptive … will go beyond simple observation and data recording…”. However, authors do not provide what are the qualifications of the state that will be achieved “beyond…”. Please discuss.

We agree with the reviewer, In response:

We added in Study Design section: “The qualitative approach will provide a deeper understanding of the healthcare transition experiences faced by newcomer international students and their families, focusing on the meanings and views of the participants[32].”

We added in Study Design section: “This involves deeply engaging with the data to interpret and understand the implications and experiences of participants, thereby enhancing our understanding of how they experience healthcare transitions at both personal and community levels[31,33].”

5. Lines 198-211: Could authors please discuss the types of guarantees (in terms of the privacy and confidentiality of interview data) that are provided by Microsoft Teams.

We would like to thank the reviewer for this important question. Microsoft Teams was approved as a secured platform to conduct and transcribe the study interviews by the University of Toronto Health Sciences Research Ethics Board (REB # 44770). Interview transcripts will be stored electronically on a secure network using SharePoint on the University of Toronto server and all recordings will be permanently deleted once the analysis is complete.

We added in Data Collection Procedures section: “The generated transcripts will be reviewed, de-identified, and stored electronically on a secure network affiliated with the university. Any personal identifiers (names of individuals, locations, organizations, etc.) will be removed from the transcripts. The interview recordings will be securely stored and kept until transcription and analysis are completed. Once these processes are finalized, all the recordings will be permanently deleted to ensure participants' privacy and confidentiality are maintained in accordance with ethical research guidelines.”

6. Lines 208-210: Please discuss if authors have concerns about how comparable the interview responses will be when camera is on as opposed to when camera is off.

 We would like to thank the reviewer for raising this critical point. We want to confirm that we do not have concerns from a methodology perspective. Our study employs conventional content analysis, which primarily focuses on the data extracted from transcripts and audio recordings. This approach is distinct from discourse analysis, where non-verbal cues, such as body language, play a crucial role in data interpretation. Therefore, the comparability of interview responses is not significantly impacted by whether the camera is on or off during the interview. Thus, allowing participants the option to turn off their cameras aims to enhance their comfort without compromising the integrity and depth of the data collected for content analysis. This flexibility helps ensure that participants feel at ease during the interview, potentially leading to more open and comprehensive responses, which are vital for the richness of the data in conventional content analysis.

7. Lines 213-231: Who will conduct the interviews? Later, it is reported that there will be at least one interviewer (line 365). Please discuss if there will be other interviewers.

We would like to confirm that only one interviewer will conduct the study interviews. 

8. Lines 213-231: Will there be debriefing after interviews? Authors do discuss revising the guide later but that seems to be only tied to the data analysis. Please specify.

We would like to confirm that there will be no debriefing after the interviews. The semi-structured interview guide will only be revised and updated based on the responses to the interview questions.

To clear the raised confusion regarding updating the interview guide, we added to Data Analysis and Reporting section: “Updating the interview guide will involve only minor changes, such as including additional probes or modifying the language of some questions if they are found to be misunderstood.”

9. Lines 233-243: Will the interview guide be pilot tested?

We agree with the reviewer that the interview guide should be tested. In response:

We added to Interview Questions section: “To ensure the clarity and effectiveness of these questions, the interview guide will be pilot tested with a small group of participants (2-3). This pilot test will help refine the questions to better capture comprehensive and relevant data.”

10. Lines 234-237: “Semi-structured” interviews? Please tie back to “in-depth” interviews.

We would like to clarify that in-depth interviews will follow a semi-structured interview guide that consists of 5 open-ended main and probing questions. Thus, ‘semi-structured’ refers to the guide not the interviews. 

To address the highlighted confusion, we added in Study Design section: “In-depth interviews aim at exploring comprehensive perspectives, experiences, and motivations through detailed, open-ended interactions between the researcher and participant. Additionally, it emphasizes deep understanding by using semi-structured interview guide with probing questions that encourage participants to extensively express their thoughts and feelings [28,29].”

11. Lines 245-272: How will authors ensure that their results are credible?

We would like to thank the reviewer for pointing out this critical point. To ensure the results credibility in our study:

The findings will be presented using several quotes from different participants. This approach not only increases transparency but also allows readers to see the 

---

## [Editor Report · Decision Letter 1]

20 May 2024

Exploring the pharmacist’s role in supporting newcomer international students and their families with the transition to the Canadian healthcare system including medication use: Protocol for a qualitative study

PONE-D-24-06002R1

Dear Dr. Aboelzahab,

We’re pleased to inform you that your manuscript has been judged scientifically suitable for publication and will be formally accepted for publication once it meets all outstanding technical requirements.

Kind regards,

Kanchan Thapa, MPH, MPhil

Academic Editor

PLOS ONE

Additional Editor Comments (optional):

Dear Authors,

Regarding your sample size, approximately 15-20 or approximately 20. I understand that it's a qualitative study, and you've stated that you'll collect data up to saturation. However, I've noticed two different ways of presenting your sample. Therefore, I suggest maintaining consistency when presenting your sample size.

Its a nice piece of works which can guide other research and health care services linked with migration of international students in many developed countries including Canada. 
---

## [Editor Report · Acceptance letter]

28 May 2024

PONE-D-24-06002R1 

PLOS ONE

Dear Dr. Aboelzahab, 

I'm pleased to inform you that your manuscript has been deemed suitable for publication in PLOS ONE. Congratulations! Your manuscript is now being handed over to our production team.

Kind regards, 

on behalf of

Mr. Kanchan Thapa 

Academic Editor

PLOS ONE